# A hierarchically assembled 88-nuclei silver-thiacalix[4]arene nanocluster

Zhi Wang[1], Hai-Feng Su[2], Yi-Wen Gong[1], Qing-Ping Qu[1], Yan-Feng Bi [3], Chen-Ho Tung[1], Di Sun [1]* & Lan-Sun Zheng[2]

Thiacalix[4]arenes as a family of promising ligands have been widely used to construct polynuclear metal clusters, but scarcely employed in silver nanoclusters. Herein, an anion-templated $Ag_{88}$ nanocluster (SD/Ag88a) built from $p$-tert-butylthiacalix[4]arene ($H_4TC4A$) is reported. Single-crystal X-ray diffraction reveals that $C_4$-symmetric SD/Ag88a resembles a metal-organic super calix comprised of eight $TC4A^{4-}$ as walls and 88 silver atoms as base, which can be deconstructed to eight $[CrO_4@Ag_{11}(TC4A)(EtS)_4(OAc)]$ secondary building units arranged in an annulus encircling a $CrO_4^{2-}$ in the center. Local and global anion template effects from chromates are individually manifested in SD/Ag88a. The solution stability and hierarchical assembly mechanism of SD/Ag88a are studied by using electrospray mass spectrometry. The $Ag_{88}$ nanocluster represents the highest nuclearity metal cluster capped by $TC4A^{4-}$. This work not only exemplify the specific macrocyclic effects of $TC4A^{4-}$ in the construction of silver nanocluster but also realize the shape heredity of $TC4A^{4-}$ to overall silver super calix.

[1] Key Laboratory of Colloid and Interface Chemistry, Ministry of Education, School of Chemistry and Chemical Engineering, State Key Laboratory of Crystal Materials, Shandong University, Jinan 250100, People's Republic of China. [2] State Key Laboratory for Physical Chemistry of Solid Surfaces and Department of Chemistry, College of Chemistry and Chemical Engineering, Xiamen University, Xiamen 361005, People's Republic of China. [3] College of Chemistry, Chemical Engineering and Environmental Engineering, Liaoning Shihua University, Fushun 113001, People's Republic of China. *email: dsun@sdu.edu.cn

Due to the esthetic structures and a plethora of promising properties, silver nanoclusters have emerged as a hot topic garnering great interests over the last decade.[1–12] However, their synthetic chemistry is still in the embryo, and trial and error is now the most popular synthetic routine. Regarding their assembly, protecting ligand is one of the most important pre-requisites we must consider, and the widely recognized candidates are thiols, alkynes, and phosphines, or their combinations.[13–15] Later, the advancements of anion template strategy[16–19] and geometric polyhedral principle[20–22] promote the rational design and synthesis of silver nanoclusters to a higher level. Compared with the above-mentioned organic ligands bearing single-coordination site, macrocyclic ligands with multiple pre-organized coordination sites are much more desired in the construction of silver nanoclusters because of their reinforcement effect originating from the cooperative coordination of multiple binding groups. These considerations are reminiscent of "third generation" calixarenes–thiacalix[4]arenes, which are macrocyclic tetramers of phenols joined by sulfur atoms[23] and have been recognized as a family of good ligands in the assembly of poly-nuclear metal clusters and cages.[24–27] Based on the multiple coordination sites of –OH and –S– groups on them, a series of large metal nanoclusters or nanocages, including $Co_{16}$, $Co_{24}$, $Mn_{24}$, $Co_{32}$, $Ni_{32}$, and $Ni_{40}$, have been reported by Liao and Hong groups.[28–33] However, thiacalix[4]arene-protected silver nanocluster is still rudimentary, and only two closely related $p$-tert-butylthiacalix[4]arene ($H_4TC4A$)-capped reductive $Ag_{34}$ and $Ag_{35}$ nanoclusters have been reported.[34,35] Although the coordination sites of $H_4TC4A$ favor to support silver nanoclusters, the bulky skeleton of $H_4TC4A$ also brings a big challenge in growth of single crystals, which is very crucial to understand the structural details of both metal core and metal–ligand interface.

Except for the ligand selection, oxoanion template plays another dominant role in constructing silver nanoclusters due to the strong directional effect arising from Ag–O interaction.[36–41] In most cases of anion-templated assembly, oxoanions commonly exert the global effect that means the metal ions aggregate around them in a roughly chaotic fashion without any precedence. As we know, shuttlecock-like $\{M_4(TC4A)\}$ (M = Mn, Fe, Co, and Ni) is a very common secondary building unit (SBU) in $TC4A^{4-}$-capped metal nanoclusters;[42] however, we do not know what is the SBU if metal is switched to silver in the presence of the anion template. More importantly, we are also unclear that whether the as-formed $\{(template)@Ag_x(TC4A)\}$ SBUs can be further reorganized around the oxoanion template again to form the hierarchical motif. Thus, correlating the SBUs and the final structure of $TC4A^{4-}$-capped silver nanocluster is very important for understanding their syntheses and assembly mechanism.

Considering the rich advantages of $H_4TC4A$ in coordination chemistry and the powerful anion template effect in silver nanoclusters, we are extending our researches to combine them together in the synthesis of silver nanoclusters. Herein, we present a $C_4$-symmetric silver nanocluster $(K_2[(CrO_4)_9@Ag_{88}(TC4A)_8$ $(EtS)_{32}(OAc)_8]\cdot8CH_3CN\cdot4DMF$; **SD/Ag88a**) with a super calix shape containing the $[CrO_4@Ag_{11}(TC4A)(EtS)_4(OAc)]$ as SBU. Eight SBUs are further cyclized into an $Ag_{88}$ cluster around a central $CrO_4^{2-}$. This silver nanocluster is the highest-nuclearity metal cluster capped by $TC4A^{4-}$. The structural features including the special ligand effect, local and global anion template effects, as well as the hierarchical assembly in **SD/Ag88a** will be discussed in detail.

## Results

### Structures of SD/Ag88a and SD/Ag88b.

Briefly, **SD/Ag88a** was facilely prepared by the reaction of $(EtSAg)_n$, $H_4TC4A$, AgOAc,

**Fig. 1 Synthetic route for super calix of SD/Ag88a.** DCM = dichloromethane, DMF = $N,N$-dimethylformamide. Color legends for objects: red: $H_4TC4A$ ligand; green: super calix; pink: silver atom; blue: $CrO_4^{2-}$; yellow: the base of super calix.

and $K_2Cr_2O_7$ in the mixed solvent system containing acetonitrile, dichloromethane, and DMF at room temperature (Fig. 1). The red prism crystals can be crystallized after 2 weeks and collected together by filtration as bulk samples (~10%). The higher-yield synthesis of **SD/Ag88a** can be achieved using solvothermal reaction at 65 °C (~40%). The AgOAc in the synthesis of **SD/Ag88a** is very crucial because we have tried the other eight different silver salts available in our laboratory, including $AgBF_4$, $CF_3COOAg$, $CH_3SO_3Ag$, $CF_3SO_3Ag$, $AgNO_3$, $AgSbF_6$, $PhCOOAg$, and $p$-TOSAg, but none of them can produce **SD/Ag88a**. Auxiliary $EtS^-$ ligand also shows steric hindrance-related influence on the formation of **SD/Ag88a** because other larger alkylthiols such as $^tBuSH$ or $^iPrSH$ cannot produce **SD/Ag88a** under the same assembly condition. Of note, by mixing AgOAc with $AgSbF_6$ in this system, we can isolate a similar $Ag_{88}$ cluster $(K_2[(CrO_4)_9@Ag_{88}(TC4A)_8(EtS)_{32}(OAc)_8(CH_3CN)]\cdot8CH_3CN$; **SD/Ag88b**), but crystallized in monoclinic $P2_1/c$ space group. The detailed structure diagrams for **SD/Ag88b** are shown in Supplementary Fig. 1. A series of characterization techniques such as single-crystal X-ray diffraction (SCXRD), powder X-ray diffraction (PXRD), Fourier transform-infrared spectroscopy (FTIR), UV–Vis spectroscopy, thermogravimetric analysis (TGA), dynamic light scattering (DLS), energy-dispersive X-ray spectroscopy (EDS), and transmission electron microscopy (TEM) were used in this system (Supplementary Figs. 7–17).

X-ray diffraction analyses on single crystals (Supplementary Fig. 18) of **SD/Ag88a** and **SD/Ag88b** revealed that they crystallize in tetragonal $P4/n$ and monoclinic $P2_1/c$ space groups, respectively (Supplementary Table 1). More structural diagrams and crystallographic data plots for them are shown in Supplementary Figs. 19–29. The composition of **SD/Ag88a** was determined as $\{K_2[(CrO_4)_9@Ag_{88}(TC4A)_8(EtS)_{32}(OAc)_8]\cdot8CH_3CN\cdot4DMF\}$. The composition of **SD/Ag88b** has one more coordinated $CH_3CN$ on the surface of the cluster compared with **SD/Ag88a**. The asymmetric unit of **SD/Ag88a** contains a quarter of $Ag_{88}$ cluster and a crystallographic fourfold axis passes through Cr atom of the central $CrO_4^{2-}$, whereas no crystallographic symmetry element coincides with the $Ag_{88}$ cluster of **SD/Ag88b**, so a complete molecule was observed in the asymmetric unit. As a result, the overall 88-silver metallic framework of **SD/Ag88b** is more distorted than that of **SD/Ag88a**.

Due to the structural similarity between **SD/Ag88a** and **SD/Ag88b**, we just describe and discuss their structures below by taking **SD/Ag88a** as a representative. As shown in Fig. 2, **SD/Ag88a** looks like a super calix composed of 88 silver atoms, 32 $EtS^-$, 8 $TC4A^{4-}$, 8 $OAc^-$, and 9 $CrO_4^{2-}$ anions. Among them, 88 silver atoms and 8 $TC4A^{4-}$ ligands roughly constitute the base and wall of the super calix, respectively. The equator diameter and the height of **SD/Ag88a** are 2.2 and 1.1 nm, respectively, by removing the organic shell.

The metallic skeleton of 88 silver atoms can be divided into eight $CrO_4^{2-}$-templated $Ag_{11}$ SBUs and each of them is capped by one $TC4A^{4-}$ with a cone-shaped conformation to form an

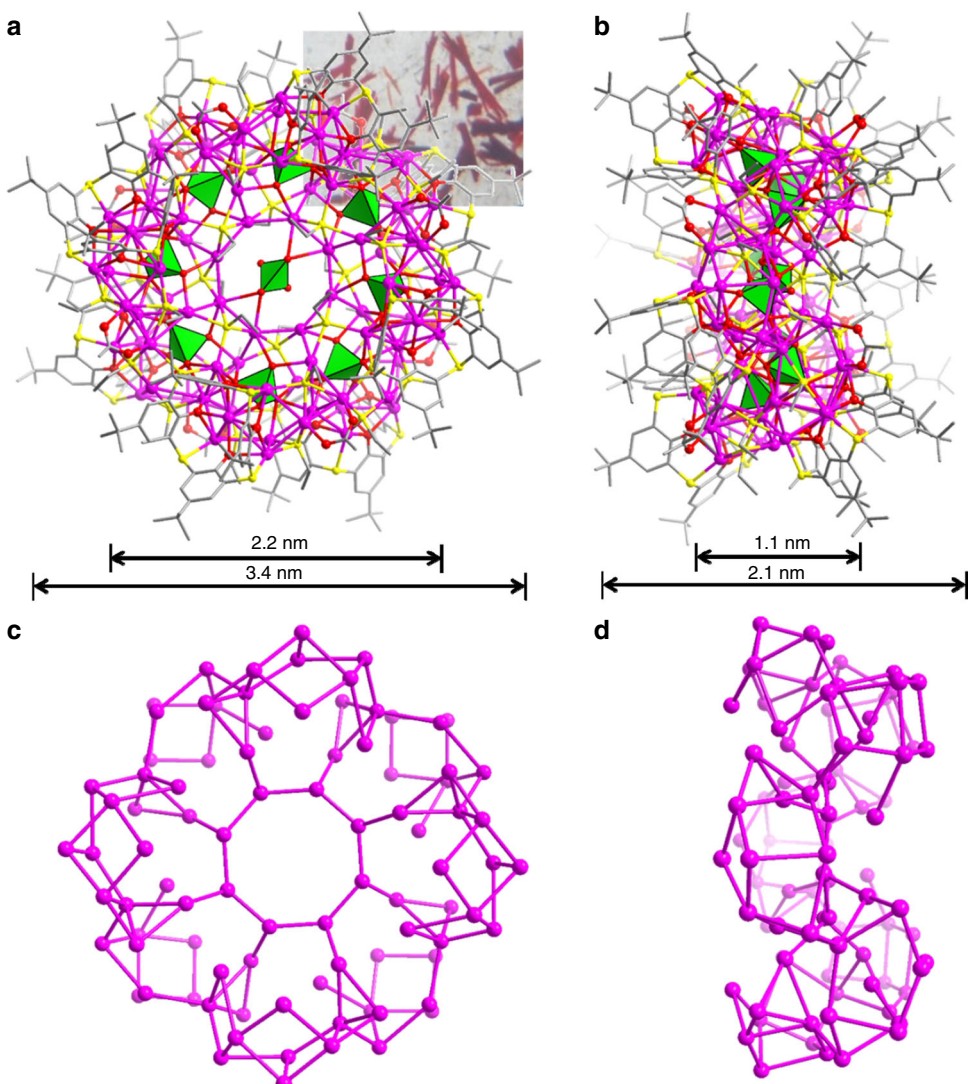

**Fig. 2 Single-crystal X-ray structure of SD/Ag88a. a** and **b** Total structures of $Ag_{88}$ super calix viewed along two orthogonal directions. The inset in Fig. 2a is the photograph of crystals of **SD/Ag88a** taken by using a digital camera under the microscope. **c** and **d** The skeletal structure of **SD/Ag88a** by removing all organic ligands and anion templates viewed along two orthogonal directions. Color labels: purple, Ag; yellow, S; gray, C; red, O; green polyhedra, $CrO_4^{2-}$.

irregular SBU with a composition of $[CrO_4@Ag_{11}(TC4A)(EtS)_4(OAc)]$ (Fig. 3a). The $CrO_4^{2-}$ plays the local templating effect in such SBU using a $\mu_{10}$-$\kappa^3$:$\kappa^3$:$\kappa^2$:$\kappa^2$ mode. In the asymmetric unit, there are two $Ag_{11}$ SBUs fused together with two $TC4A^{4-}$ ligands locating in a nearly perpendicular orientation (Supplementary Fig. 2). In each cavity of $TC4A^{4-}$, one $CH_3CN$ molecule is encapsulated and its N atom points out of the bigger opening of $TC4A^{4-}$ (Supplementary Fig. 3). The sunken voids formed after removing $CH_3CN$ molecule from $TC4A^{4-}$ are clearly shown in Supplementary Fig. 4. Two crystallographic unique $TC4A^{4-}$ ligands show different coordination modes using both phenolic hydroxyl and bridging sulfur atoms, $\mu_6$-$\kappa_o^2$:$\kappa_o^3$:$\kappa_o^3$:$\kappa_o^3$:$\kappa_s^1$:$\kappa_s^1$:$\kappa_s^2$ (Fig. 3b) and $\mu_7$-$\kappa_o^3$:$\kappa_o^3$:$\kappa_o^3$:$\kappa_o^3$:$\kappa_s^1$:$\kappa_s^1$:$\kappa_s^1$:$\kappa_s^2$ (Fig. 3c). The Ag–O and Ag–S bond lengths related to $TC4A^{4-}$ fall in the ranges of 2.255(5)−2.70(1) Å and 2.511(5)−2.753(5) Å, respectively (Supplementary Table 2). The $OAc^-$ uses bidentate bridging ($\mu_2$-$\kappa^1$:$\kappa^1$) mode to coordinate on the $Ag_{11}$ SBU (Ag–O: 2.14(2)−2.41(2) Å). Two of four $EtS^-$ ligands in each SBU adopt $\mu_4$ mode to cap on $Ag_{11}$ SBU (Ag–S: 2.397(5)−2.641(7) Å), whereas the other two (one in $\mu_3$ and another in $\mu_4$ mode) combine with $TC4A^{4-}$ bridges to consolidate the joints between

SBUs (Supplementary Fig. 5). One remaining central $CrO_4^{2-}$ anion ($\mu_4$-$\kappa^2$:$\kappa^2$:$\kappa^0$:$\kappa^0$) uses the global templating effect to organize eight $Ag_{11}$ SBUs into an annulus finally (Fig. 3d); thus the best description for **SD/Ag88a** is $\{CrO_4@[CrO_4@Ag_{11}(TC4A)(EtS)_4(OAc)]_8\}$. Although the $TC4A^{4-}$ ligands effectively cover on **SD/Ag88a**, we should not neglect the importance of auxiliary small $EtS^-$ and $OAc^-$ ligands that fill into the coordination unsaturation regions left after $TC4A^{4-}$ coverage. The argentophilic interactions, featured as the Ag···Ag distances shorter than 3.44 Å falling in the range of 2.825(2)−3.418(2) Å, reinforce the overall $Ag_{88}$ skeleton.[43–45] Although linking cationic shuttlecock-like $\{M_4(TC4A)\}$ (M = Mn, Fe, Co, and Ni) SBUs by carboxylates can form a very large nanocage with total metal counts more than 30,[29] there are no $TC4A^{4-}$-protected metal clusters with nuclearity higher than 80; thus **SD/Ag88a** is the highest-nuclearity metal cluster capped by $TC4A^{4-}$. Compared with the known biggest silver cluster, $[Ag_{490}S_{188}(StC_5H_{11})_{114}]$,[15] the 88-nuclei silver super calix represents a brand-new structure model in the silver cluster family.

The packing of **SD/Ag88a** is also quite interesting and shown in Fig. 4a, c. The $Ag_{88}$ super calix is lined in a face-to-face fashion

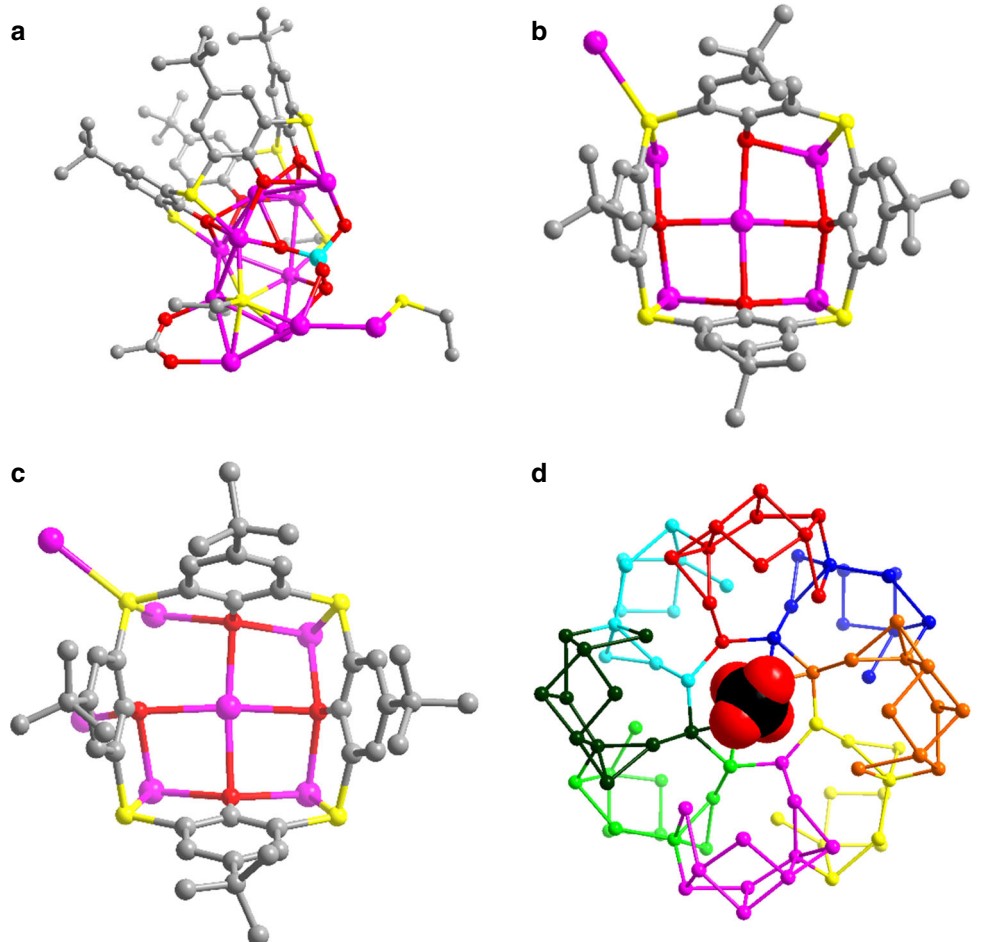

**Fig. 3 Ag₁₁ SBU in SD/Ag88a. a** The ball-and-stick mode of the structure of [CrO₄@Ag₁₁(TC4A)(EtS)₄(OAc)] SBU. **b**, **c** Two different coordination modes of TC4A⁴⁻ ligands. Color labels: purple, Ag; yellow, S; gray, C; red, O; cyan, Cr. **d** The Ag₈₈ annulus built from eight Ag₁₁ SBUs around the central CrO₄²⁻ anion. Eight Ag₁₁ SBUs are individually colored.

to form a 1D nanotube running along [001] direction. Such packing is mainly dictated by intercluster van der Waals interaction between *t*-butyl groups on the upper rims and generates some voids as shown in Supplementary Fig. 6. The distance between adjacent two **SD/Ag88a** nanoclusters is 23.14 Å based on the separation between two Cr1 atoms. The similar packing was also observed in a chiral lead metal–organic nanotube based on β-cyclodextrin.[46] Interestingly, the packing fashion of **SD/Ag88b** is completely different from that of **SD/Ag88a**. The intercluster van der Waals interactions that dominated face-to-side arrangement were found in the packing of **SD/Ag88b** (Fig. 4b, d). The centroid separation between two Ag₈₈ clusters of **SD/Ag88b** is 26.27 Å. The larger separations between clusters indicated their loose packing that may be sensitive to co-crystallized solvents, although no satisfactory structural model for all solvent positions could be determined from SCXRD analysis. The different weight losses in the first step upon heating in N₂ stream observed in the TGA curves suggested the different solvent-filling in the crystals (Supplementary Fig. 7a) and the residues after TGA are primarily metallic silver (Supplementary Fig. 7b).

**Solution behaviors of SD/Ag88a.** Mass spectrometry, DLS, and TEM were utilized to check the solution behavior of **SD/Ag88a** dissolved in CH₂Cl₂. The ESI-MS of **SD/Ag88a** contains five major isotope-distribution envelops (**1a–1e**) in the *m/z* range lower than

4000 (Fig. 5a). They are trivalent species deduced from the difference ($\Delta m/z = 0.33$) between adjacent isotopic peaks in each envelop. The most dominant envelop centered at $m/z = 3038.487$ (**1d**) can be assigned to $[(CrO_4)_4@Ag_{44}(TC4A)_4(EtS)_{16}(OAc)]^{3+}$ (Calcd. $m/z = 3038.513$), which is roughly equal to a half of **SD/Ag88a** but losing one central CrO₄²⁻, three OAc⁻ anions, and some guest solvent molecules. In other words, the species **1d** is equivalent to four fused Ag₁₁ SBUs after losing three OAc⁻ anions. Based on the isotope distributions, the envelop **1c** centered at $m/z = 3009.850$ can be assigned to $[(CrO_4)_4@Ag_{43}(TC4A)_4(EtS)_{14}(OAc)_2(CH_2Cl_2)]^{3+}$ (Calcd. $m/z = 3009.859$). Interestingly, the *m/z* spacing between **1a** and **1c**, **1b** and **1d**, and **1d** and **1e** is 55.30 or 55.96, which can be attributed to the mass of one AgOAc divided by the charge state of +3, indicating the coordination–dissociation equilibrium between them involving losing or gaining one AgOAc unit. In the *m/z* range higher than 4000, we also observed two weak but recognizable peaks centered at 4503.247 (**1f**) and 4587.207 (**1g**). After checking the spacing of adjacent isotopic peaks, we found that **1f** and **1g** are divalent species and can be assigned to $[(CrO_4)_4@Ag_{43}(TC4A)_4(EtS)_{16}(OAc)]^{2+}$ (**1f**, Calcd. $m/z = 4503.318$) and $[(CrO_4)_4@Ag_{44}(TC4A)_4(EtS)_{16}(OAc)_2]^{2+}$ (**1g**, Calcd. $m/z = 4587.277$), respectively. The detailed formulae of **1a–1g** are listed in Supplementary Table 3.

To rule out the possible fragmentation pathway of Ag₈₈ cluster occurred in ESI-MS measurement, we also used DLS and TEM to examine the original CH₂Cl₂ solution of **SD/Ag88a**. Both Supplementary Figs. 8, 9a show some particles with diameters smaller than **SD/Ag88a**, which clearly suggest that the

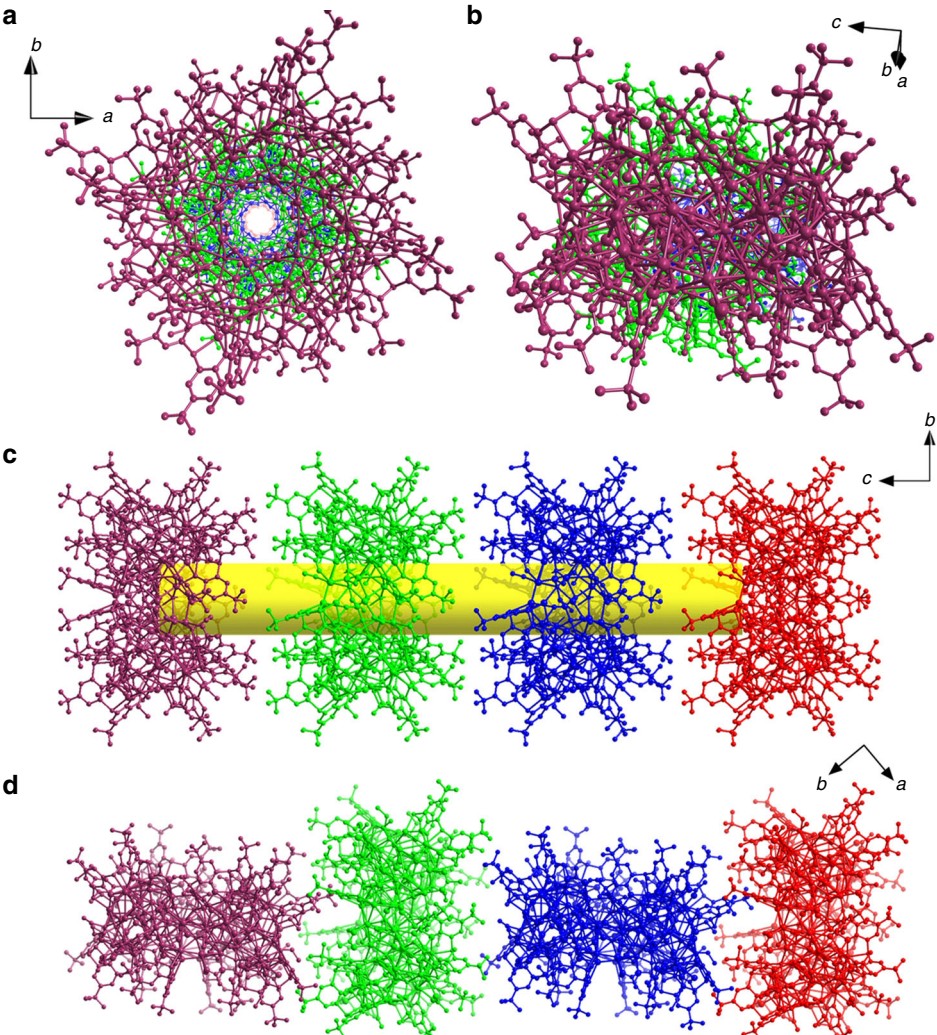

**Fig. 4 The packing of SD/Ag88a and SD/Ag88b.** Top and side views of the 1D array of **SD/Ag88a** (**a**, **c**) and **SD/Ag88b** (**b**, **d**). Different Ag$_{88}$ clusters are individually colored. The central CrO$_4{}^{2-}$ was removed for clarity.

fragmentation happens in the course of dissolution instead of the ESI-MS process. From the above-combined results, we can conclude that (i) partial **SD/Ag88a** can keep intact but mainly coexists with a half of it in the CH$_2$Cl$_2$; (ii) the coordination–dissociation equilibria involving few ligands and AgOAc exist in this system; (iii) the encapsulation of the central CrO$_4{}^{2-}$ may not happen until the final stage enclosing the overall Ag$_{88}$ super calix from two halves of Ag$_{44}$ fragments. Based on the above structural analysis and the cluster fragmentation path revealed by mass spectrometry, we can retrodict a growth route for **SD/Ag88a** from bowl-like Ag$_{11}$ SBU, a tetrameric semicircular Ag$_{44}$ fragment to the final octameric circular Ag$_{88}$ super calix (Fig. 5b).

**UV–Vis spectra and photocurrent response properties.** As shown in Fig. 6a, the solid-state UV–Vis spectra of **SD/Ag88a**, **SD/Ag88b**, and (EtSAg)$_n$ precursor were measured at 250–1000 nm at room temperature. The (EtSAg)$_n$ precursor looks yellow to the naked eye, whereas the **SD/Ag88a** and **SD/Ag88b** appear to be dark red. Both **SD/Ag88a** and **SD/Ag88b** show similar double-hump absorption profile, one narrow peak at ca. 340 nm, and one broad peak starting from ca. 370 to 800 nm. The absorption peak at 340 nm can be attributed to the n → π* transition of EtS⁻, as similarly observed in the absorption spectrum of the (EtSAg)$_n$ precursor. The broad absorption band can be attributed to the

charge transfer transition from S 3$p$ to Ag 5$s$ orbitals, which thus cause 240 and 190 nm redshifts of the absorption edges for **SD/Ag88a** and **SD/Ag88b**, respectively, compared with (EtSAg)$_n$. The bandgaps of **SD/Ag88a**, **SD/Ag88b**, and (EtSAg)$_n$ precursor were determined as 1.37, 1.48, and 2.19 eV, respectively, according to the Kubelka–Munk function (Supplementary Fig. 10), which indicates that the aggregation of silver atoms into the cluster can influence the bandgap structures that include broadening of the absorption edge and narrowing of the bandgap. In addition, both **SD/Ag88a** and **SD/Ag88b** are almost emission silent at both room temperature and liquid nitrogen temperature.

Considering the wide visible light absorption, we performed photocurrent measurements for (EtSAg)$_n$, **SD/Ag88a**, and **SD/Ag88b** in a typical three-electrode system by coating them on indium-doped SnO$_2$ (ITO) as working electrodes (platinum wire as the assisting electrode and Ag/AgCl as the reference electrode) and keeping the bias voltage at 0.6 V. The photocurrent experiments were carried out in a 0.2 M Na$_2$SO$_4$ aqueous solution under illumination upon on/off cycling irradiation with LED light (λ = 420 nm; 50 W; intervals of 10 s). Upon irradiation, photo-current density increases at 0.16, 0.12, and 0.20 μA cm$^{-2}$ for (EtSAg)$_n$, **SD/Ag88a**, and **SD/Ag88b**, respectively (Fig. 6b), which indicates that the **SD/Ag88b** possesses the best efficiency in the generation and separation of photoinduced electron/hole pairs in ITO electrodes.[47] The photocurrent density can be still

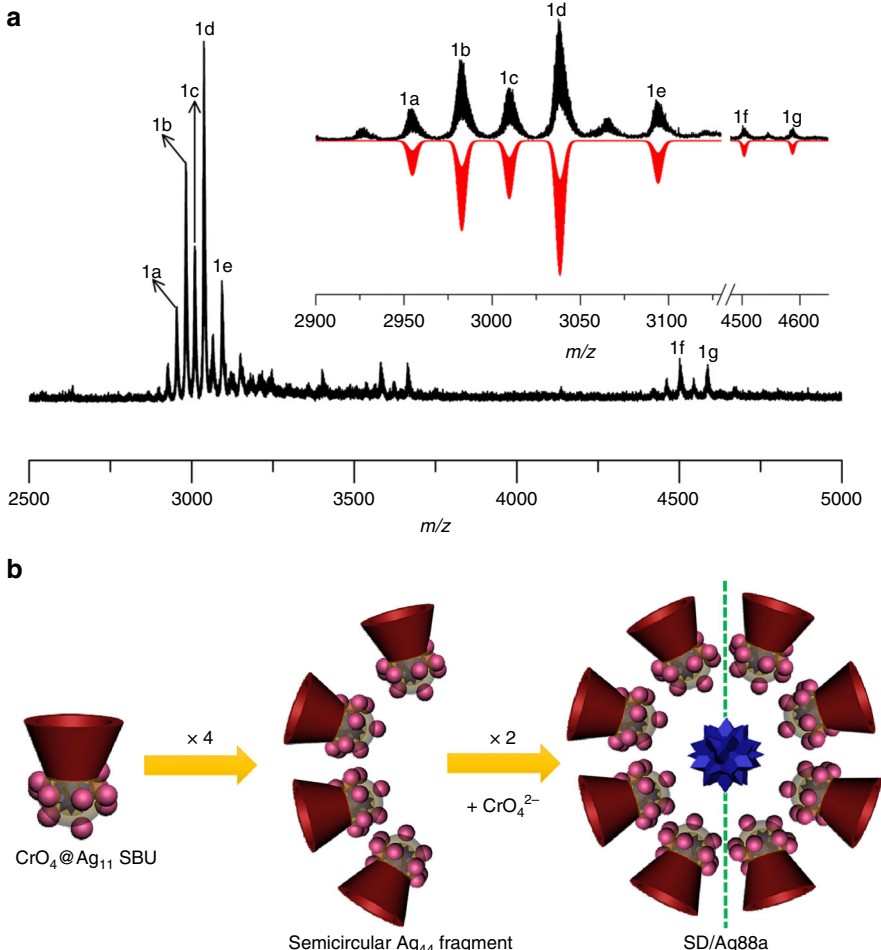

**Fig. 5 Positive-ion ESI-MS and proposed solution assembly mechanism of SD/Ag88a. a** Positive-ion ESI-MS of **SD/Ag88a** dissolved in CH$_2$Cl$_2$. Inset: the expanded experimental and simulated isotope-distribution patterns of **1a–1g**. **b** The proposed solution assembly mechanism for **SD/Ag88a**.

kept after ten on/off cycles, suggesting the response reproducibility. The generation of photocurrent may involve photoinduced charge migration from S $3p$ to the Ag $5s$ orbits.

The stability of the electrode was further proved by the compared IR spectra and PXRD patterns.[48] After the photocurrent tests, both IR spectra (Supplementary Figs. 11, 12) and PXRD patterns (Supplementary Figs. 13, 14) of samples were basically identical to those of original samples, which indicates that these samples did not undergo decomposition in the process of electrode preparation and during the photocurrent measurements.

## Discussion

In summary, we have assembled and characterized a silver-organic super calix comprising 88 silver atoms and 8 TC4A$^{4-}$ ligands. **SD/Ag88a** is the highest-nuclearity metal cluster based on TC4A$^{4-}$. Structural analysis revealed important chromate-templated Ag$_{11}$ SBUs, which are further fused into a super calix of **SD/Ag88a** with a remaining CrO$_4^{2-}$ sitting on the center. Both local and global anion-templating effects from chromates are clearly manifested in the hierarchical structure of **SD/Ag88a**. The hierarchical assembly mechanism from bowl-like Ag$_{11}$ SBU, a semicircular Ag$_{44}$ fragment to the final circular Ag$_{88}$ super calix was also revealed by using electrospray mass spectrometry (ESI-MS). The successful installation of TC4A$^{4-}$ ligand on silver nanoclusters exemplifies its powerful chelating ability and

macrocyclic effects, which surely open a bright road to assembly of silver nanoclusters using such kind of macrocyclic ligands.

## Methods

**Synthesis of SD/Ag88a.** Method A: the mixture of (EtSAg)$_n$ (0.05 mmol, 8.5 mg), H$_4$TC4A (0.015 mmol, 10.8 mg), and K$_2$Cr$_2$O$_7$ (0.025 mmol, 7.3 mg) were dissolved in mixed solvent of acetonitrile, dichloromethane, and $N,N'$-dimethylformamide (6.5 mL, v:v:v = 10:2:1). The mixed solution was stirred for 1 h at room temperature, then AgOAc (0.1 mmol, 16.7 mg) was added to the above mixture for another 3 h of stirring. The red solution was filtrated and evaporated in the dark for 2 weeks. The red prism crystals of **SD/Ag88a** were obtained in a yield of 10%.

**Synthesis of SD/Ag88a.** Method B: the mixture of (EtSAg)$_n$ (0.05 mmol, 8.5 mg), H$_4$TC4A (0.015 mmol, 10.8 mg), and K$_2$Cr$_2$O$_7$ (0.025 mmol, 7.3 mg) were dissolved in mixed solvent of acetonitrile, dichloromethane, and $N,N'$-dimethylformamide (6.5 mL, v:v:v = 10:2:1). The mixed solution was stirred for 1 h at room temperature. To this solution AgOAc (0.1 mmol, 16.7 mg) was added. The reaction continued for further 3 h of stirring, then the red mixture was sealed in a 25-mL Teflon-lined reaction vessel and kept at 65 °C for 2000 min. After cooling to room temperature, the red solution was filtrated and evaporated in the dark for 1 week. The red prism crystals of **SD/Ag88a** were isolated in a yield of 40%. Elemental analyses calc. (found) for **SD/Ag88a** (C$_{428}$H$_{588}$Ag$_{88}$Cr$_9$K$_2$N$_{12}$O$_{88}$S$_{64}$): C, 26.50 (26.51); H, 3.06 (3.08); N 0.87 (0.85)%. Selected IR peaks (cm$^{-1}$): 3382 (w), 2949 (w), 1658 (w), 1550 (w), 1434 (s), 1301 (w), 1245 (m), 1209 (w), 848 (m), 828 (s), 760 (m), 724 (m), 650 (w), 612 (w), 540 (w), 520 (w).

**Synthesis of SD/Ag88b.** The synthesis conditions were similar to those described for Method B above, but using AgOAc (0.1 mmol, 16.7 mg) and AgSbF$_6$ (0.05 mmol, 17.2 mg) instead. Red prism crystals of **SD/Ag88b** were isolated in a yield of 37%. Elemental analyses calc. (found) for **SD/Ag88b** (C$_{418}$H$_{563}$Ag$_{88}$Cr$_9$-K$_2$N$_9$O$_{84}$S$_{64}$): C, 26.22 (26.14); H, 2.96 (3.00); N 0.66 (0.59)%. Selected IR peaks

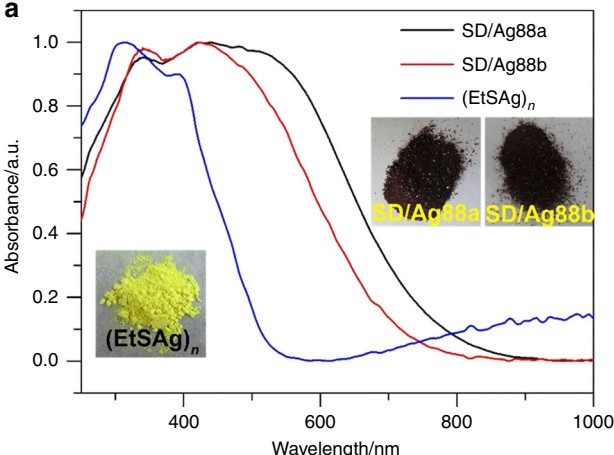

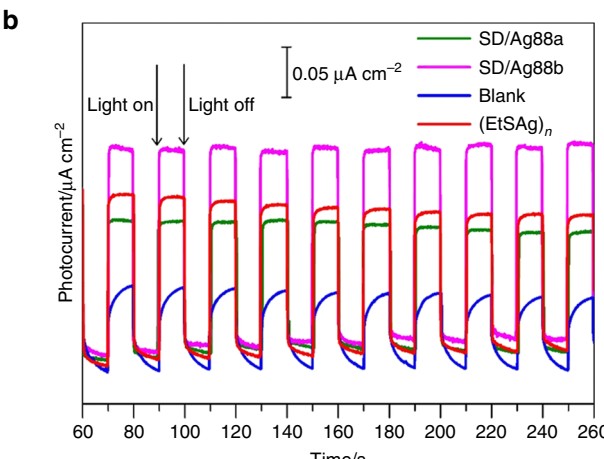

**Fig. 6 The UV–Vis spectra and photocurrent responses of SD/Ag88a and SD/Ag88b. a** The normalized UV–Vis spectra of **SD/Ag88a**, **SD/Ag88b**, and (EtSAg)$_n$ precursor in the solid state. Insets are the digital photographs of **SD/Ag88a**, **SD/Ag88b**, and (EtSAg)$_n$ taken under the ambient environment. **b** Compared photocurrent responses of blank, (EtSAg)$_n$, **SD/Ag88a**, and **SD/Ag88b** ITO electrodes in a 0.2 M Na$_2$SO$_4$ aqueous solution under repetitive irradiation.

(cm$^{-1}$): 2949 (w), 1549 (w), 1472 (w), 1433 (s), 1356 (m), 1304 (m), 1239 (m), 1051 (w), 966 (w), 855 (m), 830 (s), 764 (m), 720 (m), 648 (w), 528 (w).

## Data availability

The data that support the findings of this study are available from the corresponding author upon reasonable request. The X-ray crystallographic coordinates for structures reported in this article have been deposited at the Cambridge Crystallographic Data Centre, under deposition number CCDC: 1920453 and 1920454 for **SD/Ag88a** and **SD/Ag88b**. These data can be obtained free of charge from the Cambridge Crystallographic Data Centre via www.ccdc.cam.ac.uk/data_request/cif.

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

## Acknowledgements

This work was financially supported by the National Natural Science Foundation of China (Grant Nos. 21822107, 91961105, 21571115, and 21827801), the Natural Science Foundation of Shandong Province (Nos. JQ201803, ZR2019ZD45, and ZR2017MB061), the Taishan Scholar Project of Shandong Province of China (Nos. tsqn201812003 and ts20190908), the Qilu Youth Scholar Funding of Shandong University, and the Fundamental Research Funds of Shandong University (104.205.2.5).

## Author contributions

The original idea was conceived by D.S., experiments and data analyses were performed by Z.W., Y.-W.G., Q.-P.Q., and D.S., ESI-MS data were collected by H.-F.S., structure characterization was performed by Z.W., Y.F.B., and D.S., and the paper was drafted by D.S., Z.W., C.-H.T., and L.-S.Z. All authors have given approval to the paper.

## Competing interests

The authors declare no competing interests.
