## [Peer Review File · Nature Communications]

Reviewers' Comments:

Reviewer #1:

Remarks to the Author:

This is a quite interesting work from Sun group who reported here the first case of silver(I)-thiacalix[4]arene super calix containing up to 88 silver atoms arranged into a disc. The detailed structural elucidation of such cluster suggested that this is the largest metal cluster capped by TC4A up to now. The specific macrocyclic effects of TC4A was also reflected unprecedentedly in SD/Ag88a. I also saw the geometry inheritance of calix-like TC4A to overall silver super calix which is also very unique in ligand-protected nanoclusters. Local and global template effects from CrO₄ were separately manifested in the hierarchical structure of SD/Ag88a which also was justified using skilled ESI-MS analysis. Several important advances shown in this work are very definite in silver nanocluster field, and I believe the overall work will largely push related field go ahead. The manuscript is organized in a logical fashion and there are no scientific flaws. The scholarly presentation of the manuscript is also in line with the high standard of Nature Comm. All the characterization techniques and the conclusions derived from them have been performed in a proficient way. I am happy to recommend this paper to be accepted in Nature Comm and at the meanwhile, I would like the authors to consider the below comments and revise their paper accordingly.

- (1) Considering the synthetic difficulty, it is of curiosity for me and other readers to know the particularity of usage of EtSH. Have the authors tried to use other more common tBuSH or iPrSH in this system?
- (2) Authors told us the SD/Ag88a cannot keep intact in its CH₂Cl₂ solution and can be decompose to a half of cluster instead. However, from the ESI-MS, we can see the main fragmented peaks are around m/z 3000, but there are some weak peaks in higher m/z region (~4500) which may be some species related to Ag88. Thus I encourage the authors should assign some of them to check whether they are the parent ions of Ag88.
- (3) The coordination mode of TC4A is quite complex and may be questionable, please double check it.
- (4) Some expression of bond length is not suitable, for example, 2.411(19) Å, I suggest the authors should round s.u. in bracket to one-digit number.
- (5) The van der Waals interaction force mentioned in main text should be given as full name as 'van der Waals interaction'.

Reviewer #2:

Remarks to the Author:

The paper submitted by Sun reports the synthesis, the crystallographic characterizations and the photocurrent responses of a new Ag88 cluster capped by a thiacalix[4]arene. The growth of single crystals of big metallic cluster, large enough for crystallographic structure resolution, is still challenging. Here the use of a cyclic thiacalixarene is the key point for the new compound. Nevertheless, the paper is lacking in-depth characterizations and the photocurrents analyses should be more detailed, that's why I recommend major corrections.

- The clusters should be carefully characterized by TEM (solid state and solution) and powder XRD.
- If I am not wrong, one positive charge is missing in the formula of Ag88a and b to respect the charge balance. Is there an additional K atom?
- In the fig1, please give the meaning of the red, green, blue pink and yellow schemes with the corresponding chemical functionality.
- Authors mention that Ag88 with the highest cluster with the TC4A ligand, but please give what is the biggest silver clusters that has a known crystallographic structure with other ligands.

- In ref 23 to 33 the names of the journal are missing.
- Line 147, please precise the coordination mode of the two other EtS- ligands.
- From the TGA, the final remaining mass is different for the 2 compounds. Can you explain why?
- From ESI-MS, authors suggest that the cluster do not keep their Ag88 composition but transform to Ag44 in CH₂Cl₂ solution. Nevertheless, this fragmentation can also happen in gas phase. Please confirm that in solution the Ag88 clusters become Ag44 with UV-visible, DLS, and a microscopy technique. In addition, the formation of Ag88 from Ag44 has to be proved and the synthetic conditions should be carefully described too.
- Are those clusters soluble and stable in other solutions than CH₂Cl₂?
- The electrode should be carefully characterized too. As proved by ESI-MS, are the working species Ag88 or Ag44? Are Ag88 clusters stable in ethanol? Please add some proves of the presence of the Ag88 clusters and confirm its stability after the cycles.
- Compare the photocurrent performance of these clusters with the literature.

Reviewer #3:

Remarks to the Author:

Sun and his coworkers report two silver clusters with protecting thiacalix[4]arene and thiolate ligands. Structurally characterized large silver clusters containing thiacalix[4]arenes are rare, and the structures described in this paper are interesting, I suggest its publication after addressing a few points:

1. More ESI-MS study are necessary for confirming the cluster formula. Since the reported clusters are anionic, one will expect the MS to be measured under negative mode. Especially, three protonated water molecules are very important for charge balance, but they are not so well-defined usually with such a large metal cluster.
2. In the discussion of photocurrent response, the terms valence band and conduction band are used. I doubt if it is appropriate, because these clusters are basically molecules.
3. A very closely related paper should be cited: Chem. Sci., 2019, 10, 3360–3365.

Reviewer #1 (Remarks to the Author):

This is a quite interesting work from Sun group who reported here the first case of silver(I)-thiacalix[4]arene super calix containing up to 88 silver atoms arranged into a disc. The detailed structural elucidation of such cluster suggested that this is the largest metal cluster capped by TC4A up to now. The specific macrocyclic effects of TC4A was also reflected unprecedentedly in SD/Ag88a. I also saw the geometry inheritance of calix-like TC4A to overall silver super calix which is also very unique in ligand-protected nanoclusters. Local and global template effects from CrO₄ were separately manifested in the hierarchical structure of SD/Ag88a which also was justified using skilled ESI-MS analysis. Several important advances shown in this work are very definite in silver nanocluster field, and I believe the overall work will largely push related field go ahead. The manuscript is organized in a logical fashion and there are no scientific flaws. The scholarly presentation of the manuscript is also in line with the high standard of Nature Comm. All the characterization techniques and the conclusions derived from them have been performed in a proficient way. I am happy to recommend this paper to be accepted in Nature Comm and at the meanwhile, I would like the authors to consider the below comments and revise their paper accordingly.

Response: We are very pleased and excited by positive comments on the novelty and significance of this work. We also believe that the revised manuscript improved quality thanks to your comments.

(1) Considering the synthetic difficulty, it is of curiosity for me and other readers to know the particularity of usage of EtSH. Have the authors tried to use other more common tBuSH or iPrSH in this system?

Response: Thanks for your constructive suggestion. This is really a good suggestion regarding to the systematical studies on the synthesis chemistry of silver nanoclusters. It is obligatory for us to try different kinds of RSH ligands available in our lab including tBuSH or iPrSH in this assembly system. Unfortunately, we only got yellow solution and red precipitate after the solvothermal reaction at the same assembly condition. From the yellow solution,

no any crystalline products can be isolated. These parallel experiments demonstrates the steric hindrance of RSH ligands has an important influence on the formation of such silver super calix. For clarity, we also added one comment in main text as: Auxiliary EtS ligand also shows steric hindrance related influence on the formation of SD/Ag88a because other larger alkylthiols such as ^tBuSH or ⁱPrSH cannot produce SD/Ag88a under the same assembly condition.

(2) Authors told us the SD/Ag88a cannot keep intact in its CH₂Cl₂ solution and can be decompose to a half of cluster instead. However, from the ESI-MS, we can see the main fragmented peaks are around m/z 3000, but there are some weak peaks in higher m/z region (~4500) which may be some species related to Ag88. Thus I encourage the authors should assign some of them to check whether they are the parent ions of Ag88.

Response: Thanks for your constructive suggestion. We rechecked the ESI-MS of SD/Ag88a and observed two weak but recognizable peaks centered at 4503.247 (1f) and 4587.207 (1g) in the higher m/z region. After checking the spacing of adjacent isotopic peaks, we found 1f and 1g are divalent species and can be assigned to $[(CrO_4)_4@Ag_{43}(TC4A)_4(EtS)_{16}(OAc)]^{2+}$ (1f, Calcd. m/z = 4503.318) and $[(CrO_4)_4@Ag_{44}(TC4A)_4(EtS)_{16}(OAc)_2]^{2+}$ (1g, Calcd. m/z = 4587.277), respectively, based on matching the simulated and experimental isotopic patterns. From two species assigned above, we found that they are still not the parent ions of SD/Ag88a but fragments related to a half of SD/Ag88a. For clarity, we also added these results in main text as: In the m/z range higher than 4000, we also observed two weak but recognizable peaks centered at 4503.247 (1f) and 4587.207 (1g). After checking the spacing of adjacent isotopic peaks, we found 1f and 1g are divalent species and can be assigned to $[(CrO_4)_4@Ag_{43}(TC4A)_4(EtS)_{16}(OAc)]^{2+}$ (1f, Calcd. m/z = 4503.318) and $[(CrO_4)_4@Ag_{44}(TC4A)_4(EtS)_{16}(OAc)_2]^{2+}$ (1g, Calcd. m/z = 4587.277), respectively. Moreover, the Figure 5 was also replaced by a new one (Fig. R1 or Fig. 5a in main text) with labeled species 1f and 1g as well as their experimental and

simulated isotope-distribution patterns. The supplementary Table 3 was also revised by adding their assigned formulae accordingly (Table R1).

Figure R1

	Species	Exp.	Sim.
1a	$[(\text{CrO}_4)_4@Ag_{42}(\text{TC4A})_4(\text{EtS})_{14}(\text{OAc})(\text{CH}_2\text{Cl}_2)]^{3+}$	2953.884	2953.887
1b	$[(\text{CrO}_4)_4@Ag_{43}(\text{TC4A})_4(\text{EtS})_{16}]^{3+}$	2982.523	2982.541
1c	$[(\text{CrO}_4)_4@Ag_{43}(\text{TC4A})_4(\text{EtS})_{14}(\text{OAc})_2(\text{CH}_2\text{Cl}_2)]^{3+}$	3009.850	3009.859
1d	$[(\text{CrO}_4)_4@Ag_{44}(\text{TC4A})_4(\text{EtS})_{16}(\text{OAc})]^{3+}$	3038.487	3038.513
1e	$[(\text{CrO}_4)_4@Ag_{45}(\text{TC4A})_4(\text{EtS})_{16}(\text{OAc})_2]^{3+}$	3093.784	3093.819
1f	$[(\text{CrO}_4)_4@Ag_{43}(\text{TC4A})_4(\text{EtS})_{16}(\text{OAc})]^{2+}$	4503.247	4503.318
1g	$[(\text{CrO}_4)_4@Ag_{44}(\text{TC4A})_4(\text{EtS})_{16}(\text{OAc})_2]^{2+}$	4587.207	4587.277

Table R1

(3) The coordination mode of TC4A is quite complex and may be questionable, please double check it.

Response: Thanks for your kind reminder. Yes, TC4A⁴⁻ is a multisite and multidentate ligands and totally eight in SD/Ag88a. However, only two crystallographically unique TC4A ligands are found in the asymmetric unit. So we just need to recheck the coordination environments of two TC4A⁴⁻ ligands in this cluster. Two crystallographic unique TC4A⁴⁻ ligands show different

coordination modes using both phenolic hydroxyl and bridging sulfur atoms. Considering the potentially different deprotonated forms of H₄TC4A, we paid much attention on this issue. After carefully rechecking, two different coordination modes of TC4A⁴⁻ ligands are mainly caused by a weak Ag-O interaction (2.86 Å) that means only one phenolic hydroxyl is in a μ₂ mode and should be deprotonated, whereas all other phenolic hydroxyl groups adopt a μ₃ mode, which also clearly proves the fully deprotonated form of all H₄TC4A in SD/Ag88a. In all, we confirmed again the correctness of coordination modes of TC4A⁴⁻ in SD/Ag88a as described in main text.

(4) Some expression of bond length is not suitable, for example, 2.411(19) Å, I suggest the authors should round s.u. in bracket to one-digit number.

Response: Thanks for your careful checking. We rechecked all bond length expressions in main text and showed s.u. in bracket with one-digit number.

(5) The van der Waals interaction force mentioned in main text should be given as full name as ‘van der Waals interaction’.

Response: Thanks for your careful checking. We have revised it in the main text according to your suggestions.

Reviewer #2 (Remarks to the Author):

The paper submitted by Sun reports the synthesis, the crystallographic characterizations and the photocurrent responses of a new Ag₈₈ cluster capped by a thiacalix[4]arene. The growth of single crystals of big metallic cluster, large enough for crystallographic structure resolution, is still challenging. Here the use of a cyclic thiacalixarene is the key point for the new compound. Nevertheless, the paper is lacking in-depth characterizations and the photocurrents analyses should be more detailed, that's why I recommend major corrections.

Response: Thanks for your positive comments on our work in the growth of high-quality single crystals of big metallic cluster using of a cyclic thiacalixarene. We would also like to thank the reviewer for his/her inspiring and constructive comments/suggestions, which have been taken into careful consideration in this revision. Moreover, we added two compared photocurrents results into the corresponding section to help us to get deep insights about the photocurrent response of this kind of molecules. We believe that the revised manuscript improved quality thanks to your constructive suggestions.

- The clusters should be carefully characterized by TEM (solid state and solution) and powder XRD.

Response: Thanks for your constructive suggestions. We performed all these additional experiments as you suggested to improve the quality of this manuscript, although some of them can only provide very few useful information related to Ag₈₈ cluster.

Figure R2

- (1) From the PXRD, we can see the high purity of the bulk samples of SD/Ag88a and SD/Ag88b (see Fig. R2 or Supplementary Fig. 13 and 14). We added these results into Supplementary Information as Supplementary Fig. 13 and 14.
- (2) We tried our best to examine the morphology of the individual cluster by TEM with samples prepared by drop casting of dilute CH_2Cl_2 solution onto the ultrathin carbon film with a pipette. As shown in Fig. R3 or Supplementary Fig. 9a, nanoparticles exhibit average diameter of 2.6 nm, slightly larger than that obtained from SCXRD result (2.2 nm for metal framework), which indicates slight welding of Ag₈₈ clusters under the electron-beam conditions during TEM measurement. This is a very common phenomenon for metal nanostructures (Angew. Chem. Int. Ed. 2019, 58, 8139).

Figure R3

- (3) We also examined the morphology of solid sample of SD/Ag88a by dispersing the crystals in ethanol under the ultrasound condition using TEM. But the crystals of SD/Ag88a are hard to be dispersed into very small particles. As shown in Fig. R4 or Supplementary Fig. 9b, we can see roughly macroscopic

size and shape of crystals of SD/Ag88a. However, this result only provides few useful information related to Ag88 cluster, compared to SCXRD analysis. Anyway, we put all these results into Supplementary Information to improve the overall quality of the submission.

Figure R4

- If I am not wrong, one positive charge is missing in the formula of Ag88a and b to respect the charge balance. Is there an additional K atom?

Response: Thanks for your careful checking and kind reminder. Yes, we agreed with your opinion about the charge state of SD/Ag88a and SD/Ag88b. There should be the K^+ to balance the additional negative charge on the Ag₈₈ clusters.

Considering the potentially different deprotonated forms of H₄TC4A, we also paid much attention on this issue. After carefully rechecking, two different coordination modes of TC4A⁴⁻ ligands are mainly caused by a weak Ag-O interaction (2.86 Å) that means only one phenolic hydroxyl is in a μ_2 mode and should be deprotonated, whereas all other phenolic hydroxyl groups adopt a μ_3

mode, which also proves the fully deprotonated form of all H₄TC4A in SD/Ag88a. Then, we rechecked the formula of SD/Ag88a and SD/Ag88b and the charges (Z) of them can be calculated as follows: $Z = 88 (\text{Ag}^+) - 8 (\text{TC4A}^{4-}) - 32 (\text{EtS}^-) - 8 (\text{OAc}^-) - 9 (\text{CrO}_4^{2-}) = -2$. In order to keep charge neutrality, two cations are needed for them, although they are not directly resolved in crystallography due to their highly disordered state in lattice. Due to the usage of K₂Cr₂O₇ in the syntheses of SD/Ag88a and SD/Ag88b, we assigned counter-cation to be K⁺, which is also evidenced by EDS mapping (Fig. R5, R6 or Supplementary Fig. 15 and 16).

Figure R5

Figure R6

- In the fig1, please give the meaning of the red, green, blue pink and yellow schemes with the corresponding chemical functionality.

Response: Thanks for your constructive suggestion. We have added the color legends for the objects shown in Figure 1 as: Color legends for objects: Red: H_4TC4A ligand; green: super calix; pink: silver atom; blue: CrO_4^{2-} ; yellow: the base of super calix.

- Authors mention that Ag88 with the highest cluster with the TC4A ligand, but please give what is the biggest silver clusters that has a known crystallographic structure with other ligands.

Response: Thanks for your constructive suggestion. The known biggest silver cluster with crystallographic structure is $[Ag_{490}S_{188}(StC_5H_{11})_{114}]$ and we added this comment in the main text as: Compared to the known biggest silver cluster, $[Ag_{490}S_{188}(StC_5H_{11})_{114}]$,¹⁵ the 88-nuclei silver super calix represents a brand-new structure model in the silver cluster family.

- In ref 23 to 33 the names of the journal are missing.

Response: Thanks for your careful checking. I am sorry for making this mistake. We have added all journal names for refs 23-33.

- Line 147, please precise the coordination mode of the two other EtS- ligands.

Response: Thanks for your constructive suggestion. We rechecked the coordination modes for the other two EtS⁻ ligands between adjacent Ag₁₁ SBUs. The details about their coordination fashions were also added into main text as: whereas the other two (one in μ_3 and another in μ_4 mode) combine with $TC4A^+$ bridges to consolidate the joints between SBUs.

- From the TGA, the final remaining mass is different for the 2 compounds. Can you explain why?

Figure R7

Response: Thanks for your constructive question. Firstly, we proved the residues after TG analysis for SD/Ag88a and SD/Ag88b are primarily metallic silver by PXRD (Fig. R7 or Supplementary Fig. 7b). Although the formulae of the main structure of SD/Ag88a and SD/Ag88b are almost the same, there are different mass percentages of co-crystallized solvents due to the different cluster packing fashions in the unit cells, which will produce the different mass percentages of remaining Ag as the final product after heating at high temperature. Such explanation was also shown in the end of the paragraph before the ESI-MS section. I think aforementioned explanations can sufficiently address the concerns from this reviewer about the TGA curves.

- From ESI-MS, authors suggest that the cluster do not keep their Ag₈₈ composition but transform to Ag₄₄ in CH₂Cl₂ solution. Nevertheless, this fragmentation can also happen in gas phase. Please confirm that in solution the Ag₈₈ clusters become Ag₄₄ with UV-visible, DLS, and a microscopy technique. In addition, the formation of Ag₈₈ from Ag₄₄ has to be proved and the synthetic conditions should be carefully described too.

Response: This reviewer made very constructive comments on the solution behavior of Ag₈₈ cluster. Frankly, investigation of the real solution behavior of complex coordination clusters by mass spectrometry (MS) presents great difficulty (J. Mass Spectrom. 2003, 38, 473). It is really true that Ag₈₈ cluster possibly break up into small fragments under the ESI-MS conditions instead of in the solution. So this is a quite important issue needed to be verified regarding

when and where the fragments are formed. We performed a series experiments as suggested by this reviewer such as TEM and DLS on SD/Ag88a as a representative in this section to unambiguously justify the Ag₈₈ clusters really become some small fragments such as Ag₄₄ before injecting into the ESI-MS.

(1) Dynamic light scattering (DLS) measurement showed that the size of Ag₈₈ cluster in dilute solution has a wide distribution from 1.5 nm to 5 nm (Fig. R8 or Supplementary Fig. 8), whereas the size of Ag₈₈ cluster revealed by SCXRD is 3.4 nm, thus we believe that there are both fragments and larger aggregates coexisted with Ag₈₈ cluster when dissolved in CH₂Cl₂.

(2) The UV-Vis spectra results are shown in Fig. R9 or Supplementary Fig. 17. The CH₂Cl₂ solution of SD/Ag88a was monitored by UV-Vis spectra. From the time-course UV-Vis spectra (0-36 h) we cannot extract useful information to confirm the Ag₈₈ clusters become Ag₄₄ in solution due to the lack of spectra of the standard products, but this result can tell us the stability of the solution at the ambient environment, so we added this result into supplementary information.

Figure R9

(3) This issue is similar to the first question raised by this reviewer. We examined the morphology of the individual cluster by TEM with samples prepared by drop casting of dilute CH_2Cl_2 solution onto the ultrathin carbon film with a pipette. As shown in Supplementary Fig. 9 or Fig. R3, most of particles exhibit average diameter of 2.6 nm, slightly larger than that obtained from SCXRD result (2.2 nm for metal core), which indicates some welding of Ag_{88} clusters under the electron-beam conditions during TEM measurement. This is a very common phenomenon for metal nanostructures (Angew. Chem. Int. Ed. 2019, 58, 8139). Of note, we also should not neglect some smaller particles in TEM image, representing some fragment like Ag_{44} in TEM image. This result also proves that the Ag_{88} clusters really partially become some small fragments such as Ag_{44} before injecting into the ESI-MS.

We also totally agree with the reviewer that the direct experimental evidence for the formation of Ag_{88} from Ag_{44} is important. However, because of the highly active nature of the assembly intermediates, it is very difficult to experimentally capture and characterize their structures and further transform them into Ag_{88} , which is also beyond the scope of the present work. But it is definitely an important follow-up research topic to be pursued with significant efforts. On the

other hand, we believe the additional DLS and TEM results that we have conducted in this revision could also address this issue.

- Are those clusters soluble and stable in other solutions than CH_2Cl_2 ?

Response: Thanks for your constructive suggestion. We have tried to dissolve Ag_{88} in methanol, ethanol, acetonitrile, and toluene but none of them can dissolve Ag_{88} as well as CH_2Cl_2 .

- The electrode should be carefully characterized too. As proved by ESI-MS, are the working species Ag_{88} or Ag_{44} ? Are Ag_{88} clusters stable in ethanol? Please add some proves of the presence of the Ag_{88} clusters and confirm its stability after the cycles.

Response: Thanks for your constructive suggestions. Yes, Ag_{88} can be partially broken down to Ag_{44} in CH_2Cl_2 . But we selected ethanol to disperse sample to prepare electrode because ethanol can't dissolved Ag_{88} . Thus we believe the genuine working species on the electrode is Ag_{88} . On the other hand, the blender of Ag_{88} with naphthol will become a film after dried at room temperature which also prevented the decompose of Ag_{88} during the photocurrent measurements.

Figure R10

To further justify above discussions, we performed compared IR and PXRD measurements (Fig. R10 or Supplementary Fig. 11-14), which are the most convenient ways to prove the stability of electrode (Inorg. Chem. 2019, 58, 6312). We collected the compared IR spectra for the original sample, blender of sample with naphthol and these after photocurrent tests. As shown in Supplementary Fig. 11 and 12, the IR spectra of the samples after the photocurrent tests were basically identical to the original samples, which indicates that the samples did not undergo decomposition in the process of electrode preparation and during the photocurrent measurements. Similarly, we exposed the sample to the 420 nm light irradiation and collected compared PXRD patterns before and after irradiation to further prove the stability of the sample under the photocurrent measurement condition (Supplementary Fig. 13 and 14). We also added such comments in main text as: The stability of electrode was further proved by the compared IR spectra and PXRD patterns.⁴⁸ After the photocurrent tests both IR spectra (Supplementary Fig. 11 and 12) and PXRD patterns (Supplementary Fig. 13 and 14) of samples were basically identical to those of original samples, which indicates that these samples did not undergo decomposition in the process of electrode preparation and during the photocurrent measurements.

- Compare the photocurrent performance of these clusters with the literature.

Response: Thanks for your constructive suggestion. To the best of our knowledge, this is the first silver cluster that shows the photocurrent activity under the illumination condition, which is also one of shining points of this work. So we have no way to compare current results with previously reported examples. Anyway, it is really very important to accumulate related data for silver cluster family in the follow-up research with significant efforts. Alternatively, we measured the photocurrent activity for precursor (EtSAg)_n and compared the photocurrent performances in revised main text for them (Fig. R11 or Fig. 6b). The related comments and reference were also added into main text as: Upon irradiation, photocurrent density increase 0.16, 0.12 and 0.20 $\mu\text{A}/\text{cm}^2$ for (EtSAg)_n, SD/Ag88a and SD/Ag88b, respectively (Fig. 6b), which indicate the SD/Ag88b

possesses the best efficiency in the generation and separation of photoinduced electron/hole pairs in ITO electrodes.⁴⁷

Figure R11

Furthermore, as requested by the reviewer, “the photocurrents analyses should be more detailed”, we added two compared photocurrents results into the corresponding section to help us to get deep insights about the photocurrent response of this kind of molecules. The related comments and reference were also added into main text as: Upon irradiation, photocurrent density sharply increase 0.16, 0.12 and 0.20 $\mu\text{A}/\text{cm}^2$ for (EtSAg)_n, SD/Ag88a and SD/Ag88b, respectively (Fig. 6b), which indicate the SD/Ag88b possesses the best efficiency in the generation and separation of photoinduced electron/hole pairs in ITO electrodes.⁴⁷

Reviewer #3 (Remarks to the Author):

Sun and his coworkers report two silver clusters with protecting thiacalix[4]arene and thiolate ligands. Structurally characterized large silver clusters containing thiacalix[4]arenes are rare, and the structures described in this paper are interesting, I suggest its publication after addressing a few points:

Response: We are pleased and excited by the reviewer's positive acknowledgement on the novelty and significance of our study. We would also like to thank the reviewer for his/her inspiring and constructive comments/suggestions, which have been taken into careful consideration in this revision. We believe that the revised manuscript improved quality thanks to your constructive suggestions.

1. More ESI-MS study are necessary for confirming the cluster formula. Since the reported clusters are anionic, one will expect the MS to be measured under negative mode. Especially, three protonated water molecules are very important for charge balance, but they are not so well-defined usually with such a large metal cluster.

Response: Thanks for your constructive suggestion. Yes, we agreed with your suggestion about the measurement of ESI-MS in negative mode. In fact, both positive and negative mode ESI-MS were measured at the same time but the latter didn't give any signals which may be caused by the binding of some cations in the solution that produce neutral species, thus giving silent signals. But in positive mode, the fragmented cluster carries positive charges, so the signals can be only detected in the positive mode of ESI-MS.

Regarding the counter-cation and charge state of SD/Ag88a, we firstly rechecked the deprotonated forms of H₄TC4A in SD/Ag88a. After carefully rechecking, two different coordination modes of TC4A⁴⁻ ligands are mainly caused by a weak Ag-O interaction (2.86 Å) that means only one phenolic hydroxyl is in a μ_2 mode and should be deprotonated, whereas all other phenolic hydroxyl groups adopt a μ_3 mode, which also proves the fully deprotonated form of all H₄TC4A in SD/Ag88a. Then, we rechecked the formula of SD/Ag88a and SD/Ag88b and the charges (Z) of them can be calculated as follows: $Z = 88$ (Ag⁺)

- 8 (TC4A⁴⁺) - 32 (EtS⁻)-8 (OAc⁻) - 9 (CrO₄²⁻) = -2. In order to keep charge neutrality, two cations are needed for them, although they are not directly resolved in crystallography due to their highly disordered state in lattice. We also fully agree with this reviewer that the protonated water molecules are quite impossible for such large clusters. Thanks to the reminder from reviewer 2, we remembered the usage of K₂Cr₂O₇ in the syntheses of SD/Ag88a and SD/Ag88b, thus the counter-cation should be K⁺, which is also evidenced by EDS mapping (Fig. R5, R6 or Supplementary Fig. 15 and 16). Therefore, the formula of SD/Ag88a and SD/Ag88b should be written as K₂[(CrO₄)₉@Ag₈₈(TC4A)₈(EtS)₃₂(OAc)₈]·8CH₃CN·4DMF and K₂[(CrO₄)₉@Ag₈₈(TC4A)₈(EtS)₃₂(OAc)₈(CH₃CN)]·8CH₃CN, respectively.

2. In the discussion of photocurrent response, the terms valence band and conduction band are used. I doubt if it is appropriate, because these clusters are basically molecules.

Response: Thanks for your constructive suggestion. We revised the comment about the generation of photocurrent to: *The generation of photocurrent may involve photoinduced charge migration from S 3p to the Ag 5s orbits.*

3. A very closely related paper should be cited: Chem. Sci., 2019, 10, 3360–3365.

Response: Thanks for your constructive suggestion. We cited this very important and closely related paper in the main text as ref 35. The related comment about this was also added as: *However, thiacalix[4]arenes protected silver nanocluster is still rudimentary and only two closely related p-tert-butylthiacalix[4]arene (H₄TC4A) capped reductive Ag₃₄ and Ag₃₅ nanoclusters have been reported.*^{34,35}

Reviewers' Comments:

Reviewer #1:

Remarks to the Author:

All the points raised in the previous round of review are satisfactorily addressed.

Reviewer #2:

Remarks to the Author:

The authors fully answered all my questions. The manuscript can be accepted for publication as it is.

Reviewer #3:

Remarks to the Author:

My questions have been answered properly. The manuscript is publishable now.

Reviewer #1 (Remarks to the Author):

All the points raised in the previous round of review are satisfactorily addressed.

Response: Thank you very much. We appreciate your recommendation for publication of our work in Nature Communications.

Reviewer #2 (Remarks to the Author):

The authors fully answered all my questions. The manuscript can be accepted for publication as it is.

Response: Thank you very much. We appreciate your recommendation for publication of our work in Nature Communications.

Reviewer #3 (Remarks to the Author):

My questions have been answered properly. The manuscript is publishable now.

Response: Thank you very much. We appreciate your recommendation for publication of our work in Nature Communications.